# A Conservative Memristive Chaotic System with Extreme Multistability and Its Application in Image Encryption

**DOI:** 10.3390/e25121656

**Published:** 2023-12-13

**Authors:** Jian Li, Bo Liang, Xiefu Zhang, Zhixin Yu

**Affiliations:** 1College of Mathematics and Big Data, Guizhou Education University, Guiyang 550018, China; lijian@gznc.edu.cn (J.L.); yuzhixintop@163.com (Z.Y.); 2College of Big Data and Information Engineering, Guizhou University, Guiyang 550025, China

**Keywords:** memristor, conservation system, extreme multistability, amplitude control, image encryption

## Abstract

In this work, a novel conservative memristive chaotic system is constructed based on a smooth memristor. In addition to generating multiple types of quasi-periodic trajectories within a small range of a single parameter, the amplitude of the system can be controlled by changing the initial values. Moreover, the proposed system exhibits nonlinear dynamic characteristics, involving extreme multistability behavior of isomorphic and isomeric attractors. Finally, the proposed system is implemented using STMicroelectronics 32 and applied to image encryption. The excellent encryption performance of the conservative chaotic system is proven by an average correlation coefficient of 0.0083 and an information entropy of 7.9993, which provides a reference for further research on conservative memristive chaotic systems in the field of image encryption.

## 1. Introduction

A conservative system is a special system with a conserved volume whose dissipation is zero for all state variables. Conservative systems exhibit rich dynamic properties and have periodic, chaotic, and hyperchaotic behaviors. In past studies, scholars have found excellent chaotic properties in conservative systems. For example, a new five-dimensional conservative chaotic system was constructed and hyperchaotic behavior was discovered in this system [1]. Qi et al. designed a class of conservative chaotic systems with the very largest Lyapunov exponent (LLE) values, which exhibited a high complexity [2]. Wang et al. revealed the coexistence behavior of an infinite number of periodic trajectories by constructing a new conservative chaotic system [3]. Du et al. proposed a conservative chaotic system with two offset boosting behaviors controlled by parameters and initial values [4]. These studies on chaotic characteristics show that conservative chaotic systems have good application prospects. In recent years, to explore more dynamic characteristics of chaotic systems, memristors have been widely used in chaotic systems as special nonlinear components. As a result of the influence of memristors, some special phenomena are found in memristive chaotic systems, such as extreme multistability phenomena [5,6,7,8,9,10], merger crises [10], and multitransient behaviors [8,9,10]. However, most of these memristive systems are dissipative systems, and there are few reports of complex phenomena produced by memristors combined with conservative chaotic systems, which leaves room for the study of memristive conservative chaotic systems.

The extreme multistability of chaotic systems allows for such systems to obtain many different phase trajectories under the same parameter conditions [11], which greatly increases the complexity of the systems. In recent years, some results have been achieved in the multistability of conservative chaotic systems based on memristors. A conservative chaotic system with sinusoidal memory inductors and capacitors has been reported to exhibit homogeneous multistability self-replicating topologies [12]. The hyperchaotic behavior and the coexistence of an infinite number of conservative trajectories were revealed in a new conservative memristive chaotic system [13]. Time-varying hyperchaos and multiple offset boosting were explored in a nonautonomous four-dimensional conservative memristive chaotic system [14]. The results suggest that it is feasible to study unreported properties such as amplitude control by introducing special nonlinear properties of memristors into conservative memristive chaotic systems. For the development of conservative chaotic systems, it is significant to study new conservative chaotic systems.

At present, a large number of dissipative memristive chaotic systems have been applied to image encryption and show good encryption performance. In a chaotic encryption scheme, chaotic attractors generated by dissipative systems can be reconstructed using time delay methods [15] to crack keys and encryption systems [16]. Conservative chaotic systems are more random to motion and do not form chaotic attractors, so conservative systems have the advantage of resisting key reconstruction [17]. According to the above analysis, conservative chaotic systems should show better performance in encrypted communication. However, to the authors’ knowledge, there is currently only one report of image encryption using a conservative memristor chaotic system. A conservative memristive hyperchaotic system without equilibrium points is applied to color image encryption [14]. Therefore, studying conservative chaotic systems is critical for secure communication and engineering applications.

On the basis of the above research, we further enriched the types of conservative memristive chaotic systems. The main contributions of this paper are as follows.

A new conservative memristive system is constructed using a quaternary memristor, which is physically realized with an STMicroelectronics (STM, Geneva, Switzerland) 32 digital circuit, which provides a reference for the practical application of the chaotic system.The conservative system presents extreme isomorphic and heterogeneous multistability phenomena. The amplitude control of the initial value can be realized with the isomorphic property. The heterogeneous multistability phenomena greatly enhance the complexity of the system, which provides a guarantee for the application of secure communication.The conservative memristive chaotic system is rarely applied to image encryption, and the excellent encryption performance is verified by a variety of analysis methods. This guides the application of the conservative memristive system in the field of image encryption.

The rest of this paper is organized as follows. The memristor and conservative chaotic system model are introduced in Section 2, the dynamical characteristics of which are explored in Section 3. Section 4 presents the realization of the proposed STM 32 system. Section 5 presents the application of the memristive conservative chaotic system in image encryption. Finally, Section 6 summarizes the current work.

## 2. System Description

### 2.1. Memristor Model

In this paper, a smooth memristor [18] is used, and the mathematical model is shown in Equation (1).
(1)$$\left\{\begin{array}{l} i=W(\varphi )v=(\gamma +\varepsilon \varphi^{2}-\eta \varphi^{4})v\\ \frac{d\varphi }{dt}=v \end{array}\right.,$$
where W(φ) is the memductance function of the memristor; γ=1, E=0.25, and η=0.002 are the constant parameters of the memristor; and *i* and *v* denote the output current and voltage of the memristor. The amplitude of the voltage was set to 4 V to obtain a hysteresis curve as shown in Figure 1. It can be noted that the hysteresis curve of the memristor resembles an inclined number “8”. As the frequency increases, the sidelobes of the hysteresis curve gradually decrease, which is consistent with the characteristics of an ideal memristor.

### 2.2. System Model

A four-dimensional conservative chaotic system was reported in [2]. The mathematical model is shown in the following system (2).
(2)$$\left\{\begin{array}{l} \dot{x}=ay\\ \dot{y}=b(x^{2}+1)w-ax\\ \dot{z}=cw\\ \dot{w}=-b(x^{2}+1)y-cz \end{array}\right.,$$
where *a*, *b*, and *c* are constant parameters of the conservative system and *x*, *y*, *z,* and *w* are state variables. When parameter *a* is set to 1, the first state variable of the system (2) is the same as the internal state variable of the memristor; this means that the introduction of the memristor does not lead to an increase in the dimensions of the system equation. The (*x*^2^ + 1) term in the fourth dimension of system (2) is changed to the memristor memductance function *W*(*x*). Then, a new four-dimensional memristive conservative chaotic system is constructed. To explore richer dynamic properties, the coefficient of *x* still uses parameter *a*. The mathematical model is shown in the following system (3).
(3)$$\left\{\begin{array}{l} \dot{x}=y\\ \dot{y}=b(x^{2}+1)w-ax\\ \dot{z}=cw\\ \dot{w}=-b(W(x))y-cz \end{array}\right.,$$
where *a*, *b*, and *c* are the system parameters and *W(x)* = 1 + 0.25 *x*^2^ − 0.002 *x*^4^ is the memductance function of the memristor. The conservatism of the system (3) is proven by a dissipation degree of ∇·v=0. Letting the left side of system (3) equal zero, we find that the system has a unique equilibrium point *O* (0,0,0,0). To verify the stability of the equilibrium point, the Jacobian matrix is calculated as follows.
(4)$$J_{0}=\left(\begin{array}{cccc} 0 & 1 & 0 & 0\\ -a & 0 & 0 & b\\ 0 & 0 & 0 & c\\ 0 & -b & -c & 0 \end{array}\right).$$

The characteristic values of Equation (4) are calculated as $\lambda^{2}=\pm \sqrt{{\left(\frac{a+b^{2}+c^{2}}{2}\right)}^{2}-ac^{2}}-\frac{a+b^{2}+c^{2}}{2}$. When *a* > 0, the four eigenvalues are imaginary numbers, which is consistent with the characteristics of conservative systems.

## 3. Analysis of System Dynamic Characteristics

In this section, the MATLAB2018b platform is used to numerically analyze the nonlinear dynamic characteristics of system (3). The fourth-order Runge–Kutta algorithm is used to solve the equations of system (3). The sensitivity and multistability of the system are revealed by numerical analysis. In this paper, we consider the values of the Lyapunov exponents (*LEs*) to be zero, which satisfies the condition of −0.01< *LEi* < 0.01 (*i* = 1, 2, 3, 4), due to the certain errors in numerical calculations.

### 3.1. The Effect of Parameters on the Multitopology Behavior

The parameter sensitivity of chaotic systems directly affects their application. In this section, Lyapunov exponent spectra, bifurcation diagrams, and phase diagrams are used to describe the parameter sensitivity of the system (3).

#### 3.1.1. Multitopology Quasi-Periodic Behavior Dependent on the Parameter *a*

Setting the system parameters *b* = 1, *c* = 1, initial condition (*IC*) = (1, 0.1, 1, 0.5), and changing the parameter *a*, the Lyapunov exponent spectra and bifurcation diagrams were determined and are shown in Figure 2. The bifurcation diagram shows significant changes, indicating that system (3) is sensitive to parameter *a*. In addition, a wide range of quasi-periodic characteristics is shown in the bifurcation diagrams (see Figure 2a). As shown in Figure 2b, when parameter a is around 1.3, the LE curve has a positive value, which indicates the system is in a chaotic state. Some typical trajectories with the change in parameter *a* are displayed in Figure 3. 

#### 3.1.2. The Effect of Parameter c on Topology

Selecting the system parameters as *a* = 1, *b* = 1, *IC* = (1, 0.1, 1, 0.5) and changing the parameter *c*, the Lyapunov exponent spectra and bifurcation diagrams were determined and are plotted in Figure 4. As parameter *c* increases, the chaotic and quasi-periodic states are revealed in Figure 4a. In addition, a wide range of chaotic phenomena are exhibited, so the chaotic states of the system can be easily found for engineering. The Lyapunov exponent spectra are illustrated in Figure 4b. The regular and chaotic bifurcation in Figure 4a corresponds to the regions of LLE = 0 and LLE > 0 in Figure 4b, respectively. It is worth noting that the bifurcation diagram shows an quasi-periodic state, when parameter *c* is close to zero. The accuracy of the above analysis is illustrated by consistent results. The new trajectories are shown in Figure 5.

In summary, the system is sensitive to parameter changes and can produce a variety of novel quasi-periodic trajectories; such systems usually show good potential in practical applications.

### 3.2. Multistability Analysis

Multistability means that multiple trajectories coexist under the same parameters but with different initial values. Chaotic systems with multistability behavior are more easily applied in engineering. In this section, bifurcation diagrams and the basin of attraction are used to explore the multistability of system (3).

#### 3.2.1. Amplitude Control Depends on the Initial Value z(0) = u

Amplitude control increases the robustness of the system, which is of great significance for engineering. The system parameters and *IC* were set as *a* = 1.25, *b* = 1, *c* = 1, and *IC* = (0, 0, *u*, 0). The bifurcation diagram with initial value *z*(0) = *u* is shown in Figure 6a. The bifurcation diagram exhibits symmetry to *u* = 0. In addition, when *u* is less than zero, the value of *y* decreases as the value of *u* increases; and when *u* is greater than zero, the value of *y* increases with the increase in *u*, indicating the amplitude of the phase trajectories can be controlled through the initial values. To intuitively reflect the changing process of the amplitude, the phase trajectory with different *u* values is plotted and displayed in Figure 6b. The phase trajectory gradually shrinks from *u* = −1 to −0.1. It is worth noting that the amplitude regulation of the initial value is also extreme multistability behavior.

#### 3.2.2. Multistability Depends on z(0) and w(0)

In addition to the homomorphic extreme multistability mentioned above, there is also the coexistence behavior of heteromorphic trajectories. Set the parameters *a* = 1, *b* = 1, *c* = 0.81, and *IC* = (1, 0.1, *z*0, *w*0). Figure 7 exhibits the basin of attraction. Attraction domains of eight different colors are shown in the basin of attraction, indicating that at least eight trajectories coexist. The corresponding trajectories of different colors in the basin of attraction are plotted in Figure 8a–e, and heteromorphic trajectories can be found. It is worth noting that there are also some other coexistence trajectories besides the coexistence trajectories shown in the basin of attraction.

In summary, extreme multistability behavior is investigated in the proposed system. Furthermore, the phenomenon is also revealed that the amplitude of the phase trajectory can be controlled by the initial value.

## 4. Hardware Implementation Based on STM32

To verify the physical feasibility of the proposed system, in this section, STM32-based digital hardware is used to implement the system.

Analog circuits and digital circuits are two ways to achieve chaotic systems. Compared to the uncontrollable initial value of analog circuits, digital circuits have programmable characteristics, so digital circuits can realize the complex phenomena of chaotic systems. The digital circuit scheme is shown in Figure 9. A continuous chaotic system is fed into the STM32. The 32-bit main chip STM32F103C8T6 is used for data processing. The continuous signals are discretized using Euler’s algorithm as follows.
(5)$$\left\{\begin{array}{l} x(n+1)=x(n)+(y(n))\Delta T\\ y(n+1)=y(n)+(b(x^{2}(n)+1)w(n)-ax(n))\Delta T\\ z(n+1)=z(n)+(cw(n))\Delta T\\ w(n+1)=w(n)+(-by(n)(1+0.25x^{2}(n)-0.002x^{4}(n)))\Delta T \end{array}\right.,$$
where the precision parameter ∆T=0.001. A 16-bit dual channel digital to analog converter (model DAC8552) is then used to convert the iteratively generated digital signal. Finally, the converted analog signal is sent to the oscilloscope for display. With the parameters of *a* = 1, *b* = 1, and *c* = 0.81, the *y*-*w* phase trajectories with different *IC*s are shown in Figure 10. Ignoring the small error between theory and practice, it can be considered that the STM32 experimental results are consistent with the numerical simulations, which proves the physical feasibility of the proposed system. 

## 5. Application in Image Encryption

Since conservative chaotic systems are highly random and ergodic, conservative systems are more suitable for applications in the field of secure communication. Therefore, it is of great significance to investigate the application of conservative chaotic systems in image encryption.

### 5.1. NIST Test

In encrypted communication, a complete random sequence can guarantee the security of encryption. However, it is hard to decrypt with a true random sequence, so it is often used for irreversible encryption algorithms. The unpredictability and certainty of chaotic systems make the resulting pseudo-random sequences unique. Therefore, the encryption algorithm is usually designed as a chaotic sequence. The NIST test is a quantitative indicator that describes the randomness of chaotic sequences. Usually, the *p*-value is the reference value for detecting sequence randomness. If *p* > 0.01, the test is passed. Table 1 provides 15 sets of test results. All 15 tests showed that the chaotic sequence was random and had good application prospects.

### 5.2. Image Encryption Scheme 

In this paper, an image encryption algorithm for plain images is used. The core operations in this encryption scheme are scrambling and diffusion operations. The scrambling algorithm steps are as follows.

**Step 1** The values of (*m*, *n*) are calculated by Equation (6).
(6)$$\left\{\begin{array}{l} m=(U(i,j)+sum(A(Z(i,j),1\;to\;N)\mathrm{mod}M)+1\\ n=(V(i,j)+sum(A(1\;to\;M,W(i,j))\mathrm{mod}N)+1 \end{array}\right.,$$
where *i* and *j* are the pixel coordinate values; *U*, *V*, *W*, and *Z* are the random matrix generated by the chaotic sequence; and *M* and *N* are the sizes of the plain image *A*. If the condition (m=i||Z(i,j))||(n=j||W(i,j))||(Z(i,j)=i)||(W(i,j)=j) is satisfied, then the pixel point *A* (*i*, *j*) of the plain image remains unchanged; otherwise, *A* (*i*, *j*) switches positions with *A* (*m*, *n*).

**Step 2** Repeat **Step 1**, traversing all pixels *A* (*i*, *j*) of the plain image *A*.

The diffusion algorithm steps are as follows.

**Step 1** Set *i* = *M* and *j* = *N.*

**Step 2** Image *B* is diffused into *C* by Equation (7).
(7)$$\left\{\begin{array}{ll} C(i,j)=B(i,j)+Y(i,j)+r_{2}\;\mathrm{mod}\;2^{l} & i=M,\;j=N\\ C(i,j)=B(i,j)+C(i,j+1)+Y(i,j)\;\mathrm{mod}\;2^{l} & i=M,\;1\leq j<N\\ C(i,j)=B(i,j)+sum(C(i+1,1\;\mathrm{to}\;N))+Y(i,j)\;\mathrm{mod}\;2^{l} & 1\leq i<M,\;j=N \end{array}\right.,$$
where *r*_2_ is a random number key, *l* is the gray level of the image *B*, and *i* and *j* are the coordinates of the pixels. *i* and *j* are changed by the control variable method, and the change is −1. When *j* is reduced to 1, the next change will go to *N*, and *i* returns to *M*. After a complete operation, image *B* can be diffused into image *C.*

The whole process of the image encryption system is as follows.

**Step 1** The four initial values of the conservative chaotic system and two random numbers *r*_1_ and *r*_2_ were chosen as the keys *K* for the encryption scheme, *K* = (*x*0, *y*0, *z*0, *w*0, *r*_1_, *r*_2_).

**Step 2** Four pseudo-random sequences are generated through the proposed conservative chaotic system. Convert pseudo-random sequences into 6 M*N matrices X, Y, Z, W, U, and V (M and N are the length and width of the plain image).

**Step 3** A diffusion operation is performed, where a random matrix *X* is used to change the pixel values of the plain image *p*. Let us call the image after diffusion *A*.

**Step 4** A scrambling operation is performed; that is, four random matrices *Z*, *W*, *U,* and *V* are used to shuffle the position of pixels in image *A*. After the scrambling operation, the correlation between neighboring pixels is reduced. Let us call the image after scrambling *B*.

**Step 5** Repeat **Step 3** using matrix *Y* to obtain the final encrypted image *C*.

The decryption process is inverse to the encryption process. Under the condition of the same random sequence, the encrypted image can be completely restored.

### 5.3. Histogram Analysis

Encrypting images Goldhill, Lena, and Pepper took 0.698834 s, meaning that the encryption speed was 1.25 Mbps. Compared with the literature [19], this speed is within the acceptable range. The results of the histogram are shown in Figure 11. Figure 11a–c shows the plain images. The histograms corresponding to the plain images are shown in Figure 11d–f. Obvious pixel features and strong recognizability are displayed in the histograms of the plain images. The ciphered images are shown in Figure 11g–i. The histogram of the ciphered images can hide the image features well due to the uniform distribution of pixels, and the information of the image cannot be distinguished, as shown in Figure 11j–l. A histogram analysis shows that the encryption scheme based on the conservative memristive chaotic system has an excellent encryption effect, and ciphered images can resist attacks. To verify the regularity of the histogram, the numerical method of the Chi-square($\chi^{2}$) test was used [20], and the results are shown in Table 2. The test value of the plain image far exceeds the threshold value $\chi_{0.05}^{2}\left(255\right)=293.2478$, while the cipher image is less than this value. This shows that the histogram of the ciphered image is approximately a uniform distribution.

### 5.4. Correlation Analysis

In digital images, the correlation between adjacent pixels is strong. After using an ideal encryption scheme, the correlation is greatly reduced. The superiority of the encryption scheme is verified by testing the correlation between neighboring pixels in the ciphered image.

The correlation coefficient quantitatively reflects the correlation degree of adjacent pixels in images. The numerical value of the image correlation coefficient is positively correlated with the correlation. The closer the correlation coefficient is to 1, the stronger the correlation of the image. *N* pairs of neighboring pixel values are selected in the image, and their gray values are recorded with (*x_i_*, *y_i_*), where *i* = 1,2,3…n. The correlation coefficient is calculated as follows.
(8)$$\left\{\begin{array}{l} r_{xy}=\frac{\mathrm{cov}(x,y)}{\sqrt{D(x)}\sqrt{D(y)}},\\ \mathrm{cov}(x,y)=\frac{1}{N}\sum_{i=1}^{N}(x_{i}-E(x))(y_{i}-E(y)),\\ D(x)=\frac{1}{N}\sum_{i=1}^{N}{\left(x_{i}-E(x)\right)}^{2},\\ E(x)=\frac{1}{N}\sum_{i=1}^{N}x_{i}. \end{array}\right.,$$
where *x* and *y* are gray values of ciphered images; Cov (*x*, *y*) is the covariance; and *r_xy_* is the correlation coefficient. Table 3 shows the results of correlation coefficients for different encryption schemes. The correlation coefficients of the three directions of the plain image are close to 1, which is a strong correlation. However, the correlation coefficient of the encrypted image is very low and the correlation of the image is almost negligible, which is close to the ideal encryption effect. The conservative memristive chaotic system has an excellent encryption performance and can effectively prevent information leakage.

### 5.5. Information Entropy

Information entropy reflects the uncertainty of image information, and it is generally believed that the greater the entropy, the greater the uncertainty and the less visible the information. The information entropy is calculated as follows:(9)$$H=-\sum_{i=0}^{255}p(i)\log_{2}p(i)$$
where *p*(*i*) is the probability of the grayscale value *i*. For grayscale levels of 256, an image has an ideal information entropy of 8. The closer the value of the information entropy is to 8, the less likely the image is to the information and the more random the pixels. Table 4 shows the information entropy before and after encryption of the Goldhill, Lena, and Pepper images. By comparing different encryption schemes, we can see that the proposed encryption scheme is safe against entropy attacks [26].

To obtain the information distribution of different regions in the image, the analysis method of local information entropy is used. A high local information entropy is represented by a complex and random distribution of pixels in an image block. A low local information entropy is represented by a consistent distribution of pixel values. The average local information entropy is shown in Table 5. The local information entropy after encryption is close to 0, indicating that the information distribution is consistent, which can prevent information leakage.

### 5.6. Key Sensitivity Analysis

In the encryption system, the sensitivity of the key determines the security of the encryption system. Three indicators, the Number of Pixels Change Rate (NPCR), Unified Average Change Intensity (UACI), and Block Average Change Intensity (BACI), were used to quantitatively measure the sensitivity of the key. Two keys with only a 10^−13^ level difference were used to encrypt the plain image. The sensitivity of the key is reflected by the difference between the two ciphered images. Table 6 shows the calculation results and theoretical values of the three indicators. It is found that the calculation values are very close to the theoretical values, which indicates that the encryption system key is very sensitive and has an excellent security performance.

From all the above comparative analysis, it can be concluded that the image encryption scheme proposed in this paper has a good encryption performance and can resist most external attacks.

## 6. Conclusions

In this paper, a novel four-dimensional conservative memristive chaotic system with an equilibrium point is formulated. A numerical analysis reveals the proposed system is extremely sensitive to parameters, and a variety of novel chaotic and quasi-periodic trajectories can be found through small adjustments to these parameters. In addition, the introduction of memristors enhances the nonlinear dynamic characteristics of the proposed system, leading to the emergence of extreme multistability phenomena. Interestingly, the amplitude of the trajectory of the conservative memristive system can be regulated by the initial values, which simplifies the control strategy and enhances the robustness and complexity. The multistability phenomenon of the conservative system is realized by the digital circuit STM32, which proves the physical feasibility of the proposed conservative chaotic system. Finally, the application of conservative memristive chaotic systems in the field of image encryption is systematically analyzed. The excellent encryption performance is verified by histograms, a correlation analysis, and information entropy and key sensitivity analyses. This paper fills the gap for conservative memristive chaotic systems in image encryption applications and provides a significant guide for the application of conservative chaotic systems in secure communication. At present, there is relatively little research on image encryption schemes for conservative memristive chaotic systems, and further research is needed. At the same time, the security performance, anti-interference ability, and encryption speed of the image encryption algorithm are worth optimizing in future work. 

## Figures and Tables

**Figure 1 entropy-25-01656-f001:**
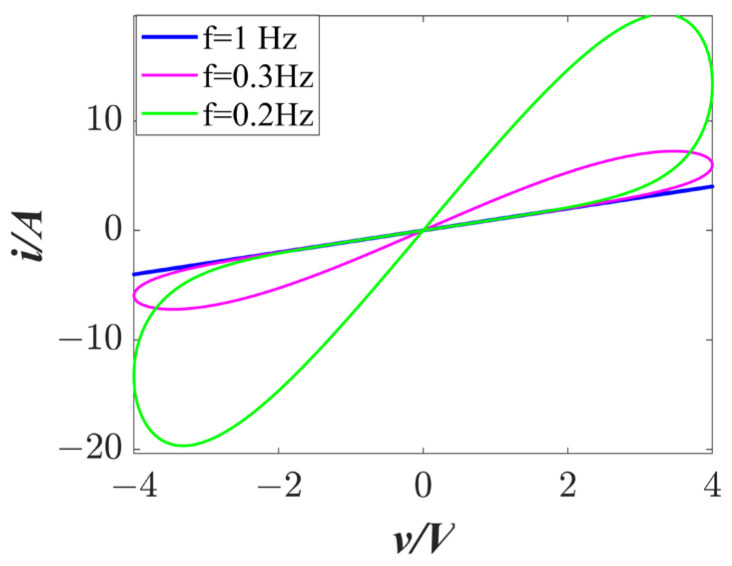
*γ* = 1, *ε* = 0.25, and *η* = 0.002. The *i–v* curve of the memristor with different frequencies (f), with an amplitude of 4 V for the sinusoidal signal input.

**Figure 2 entropy-25-01656-f002:**
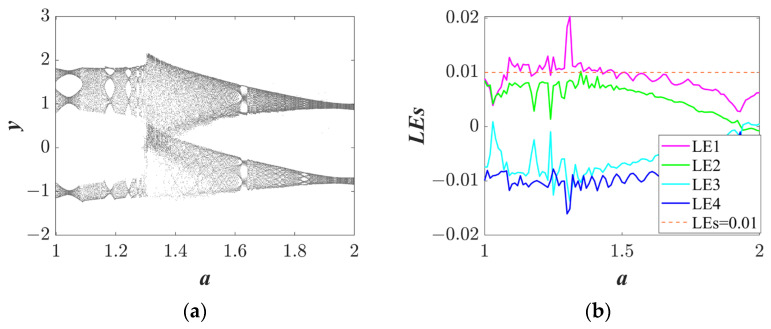
The bifurcation diagram and Lyapunov exponent spectra of system (3): (**a**) the bifurcation diagram of *a* ϵ (1, 2); and (**b**) the Lyapunov exponent spectra of *a* ϵ (1, 2).

**Figure 3 entropy-25-01656-f003:**
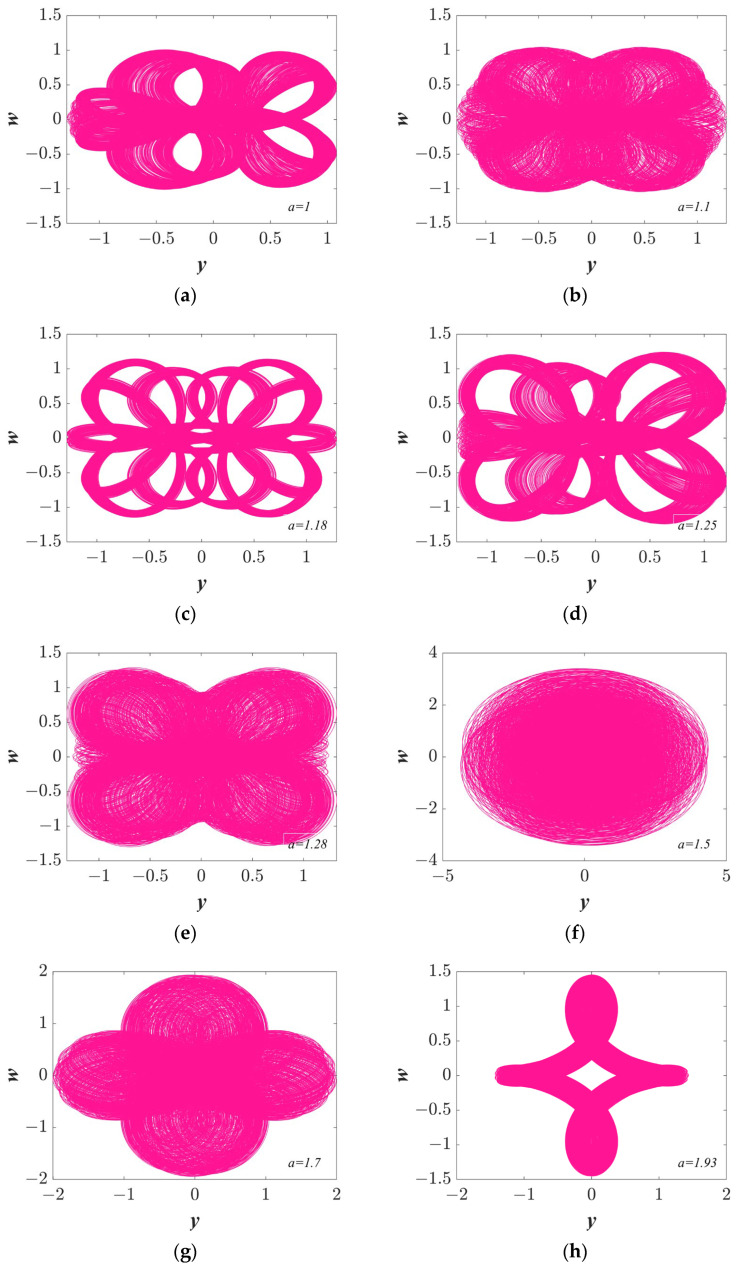
With the parameters *b* = 1, *c* = 1, and *IC* = (1, 0.1, 1, 0.5), the phase portraits vary with the parameter *a.* (**a**) *a* = 1; (**b**) *a* = 1.1; (**c**) *a* = 1.18; (**d**) *a* = 1.25; (**e**) *a* = 1.28; (**f**) *a* = 1.5; (**g**) *a* = 1.7; and (**h**) *a* = 1.93.

**Figure 4 entropy-25-01656-f004:**
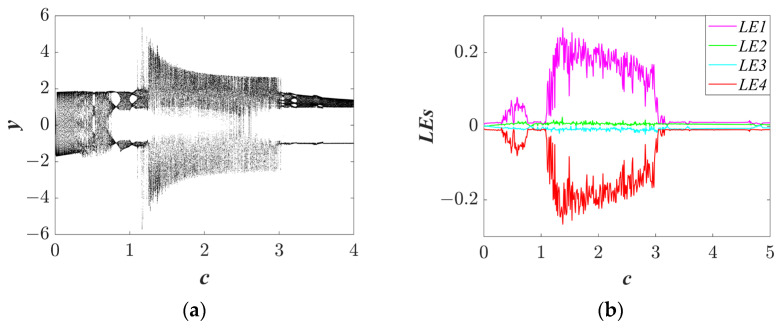
The bifurcation diagram and Lyapunov exponent spectra of system (3): (**a**) the bifurcation diagram of *c* ϵ (0, 4); and (**b**) the Lyapunov exponent spectra of *c* ϵ (0, 4).

**Figure 5 entropy-25-01656-f005:**
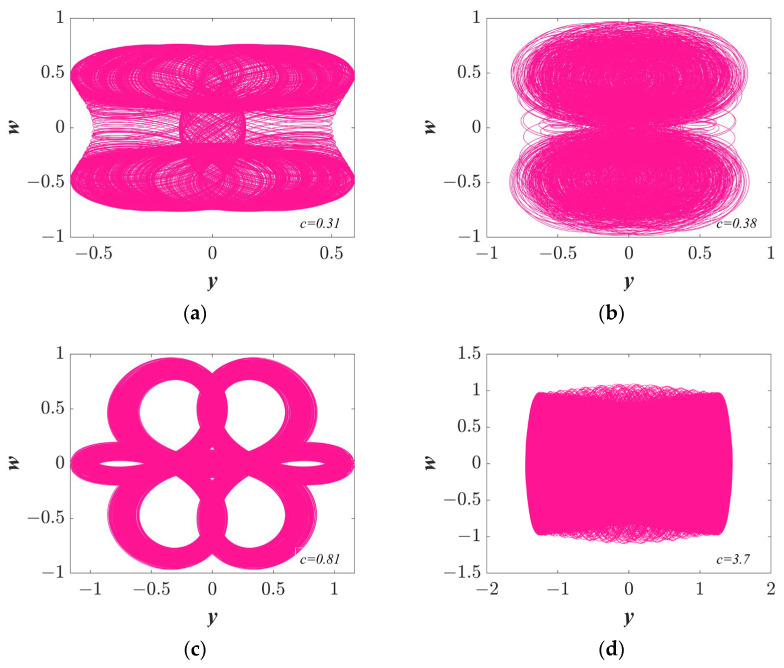
With the parameters *a* = 1, *b* = 1, and *IC* = (1, 0.1, 1, 0.5), the phase portraits vary with the parameter *c.* (**a**) *c* = 0.31; (**b**) c = 0.38; (**c**) c = 0.81; and (**d**) c = 3.7.

**Figure 6 entropy-25-01656-f006:**
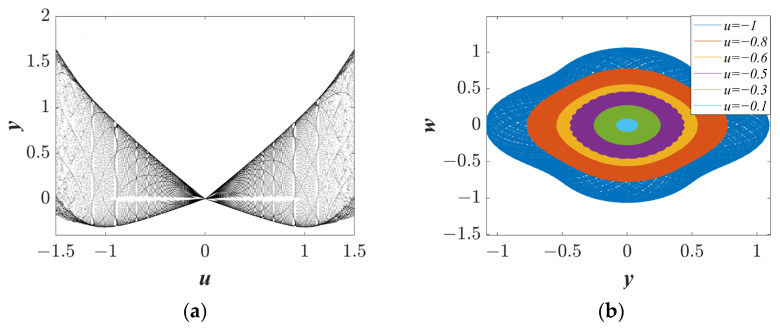
The parameters *a* = 1.25, *b* = 1, *c* = 1, and *IC* = (0, 0, *u*, 0) results in the following: (**a**) the bifurcation diagrams with initial value *z*(0) = *u*; and (**b**) the trajectories at different *u* values.

**Figure 7 entropy-25-01656-f007:**
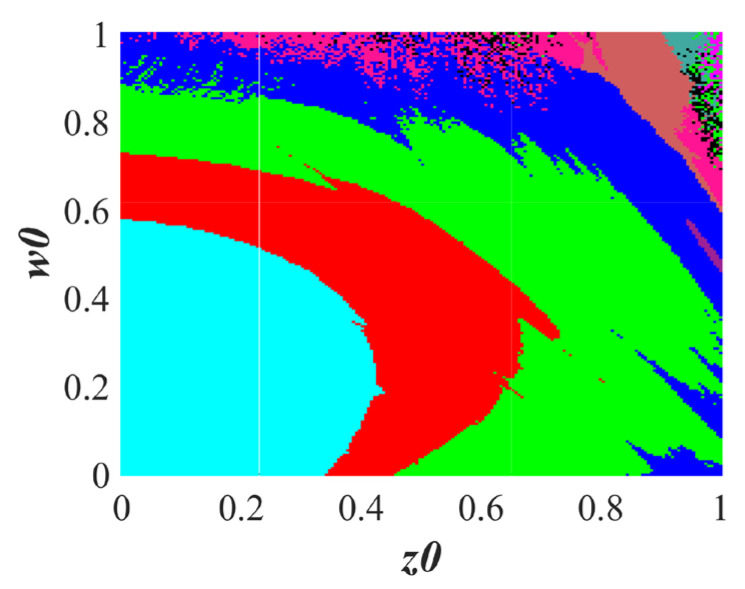
With *a* = 1, *b* = 1, and *c* = 0.81, the basin of attraction at *x0* = 1.

**Figure 8 entropy-25-01656-f008:**
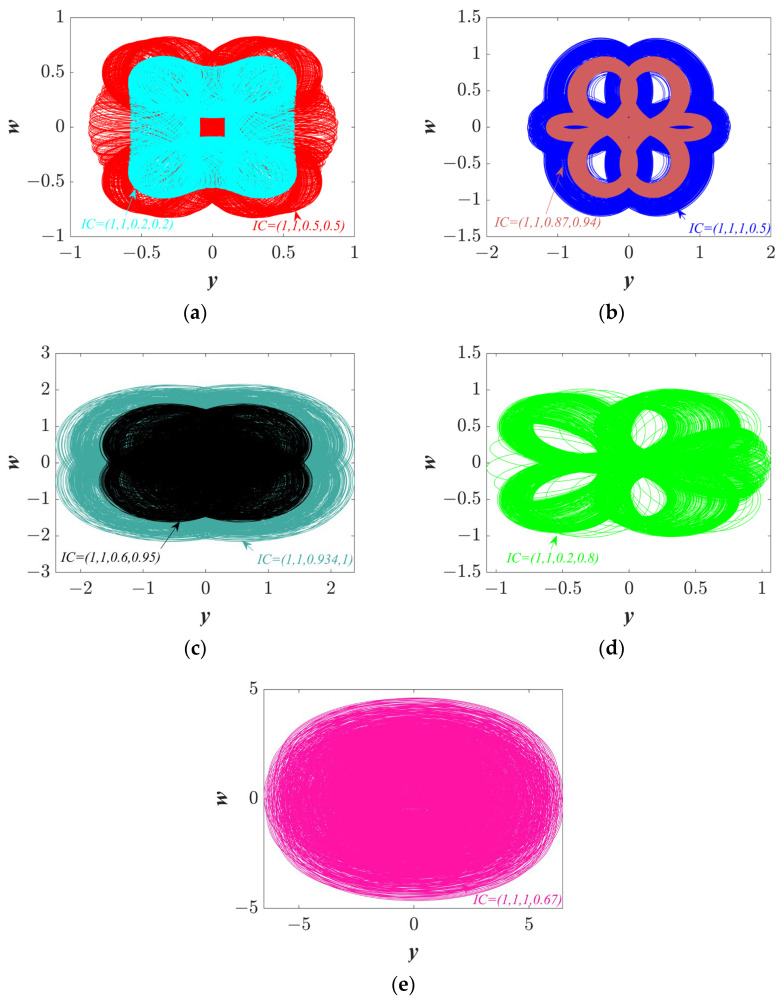
With *a* = 1, *b* = 1, and *c* = 0.81. The phase diagrams at different *ICs*: (**a**) *IC* = (1,0.1,0.1,0.2) and (1,0.1,0.5,0.5); (**b**) *IC* = (1,0.1,0.87,0.94) and (1,0.1,1,0.5); (**c**) *IC* = (1,0.1,0.6,0.95) and (1,0.1,0.934,1); (**d**) *IC* = (1,0.1,0.2,0.8); and (**e**) *IC* = (1,0.1,1,0.67).

**Figure 9 entropy-25-01656-f009:**
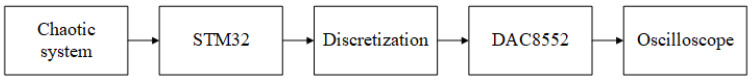
A schematic diagram of a microcontroller-based digital circuit.

**Figure 10 entropy-25-01656-f010:**
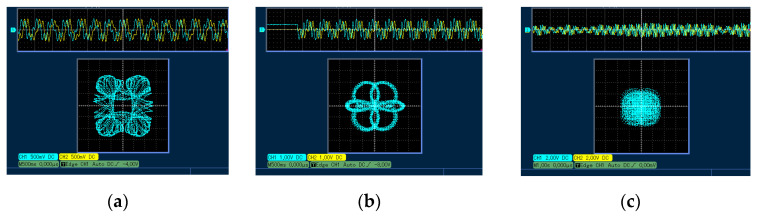
The experimental result diagrams with *a* = 1, *b* = 1, *c* = 0.81, and *IC* = (1, 0.1, *z*0, *w*0). (**a**) *z*0 = 0.2 and *w*0 = 0.2; (**b**) *z*0 = 1 and *w*0 = 0.5; and (**c**) *z*0 = 0.934 and *w*0 = 1.

**Figure 11 entropy-25-01656-f011:**
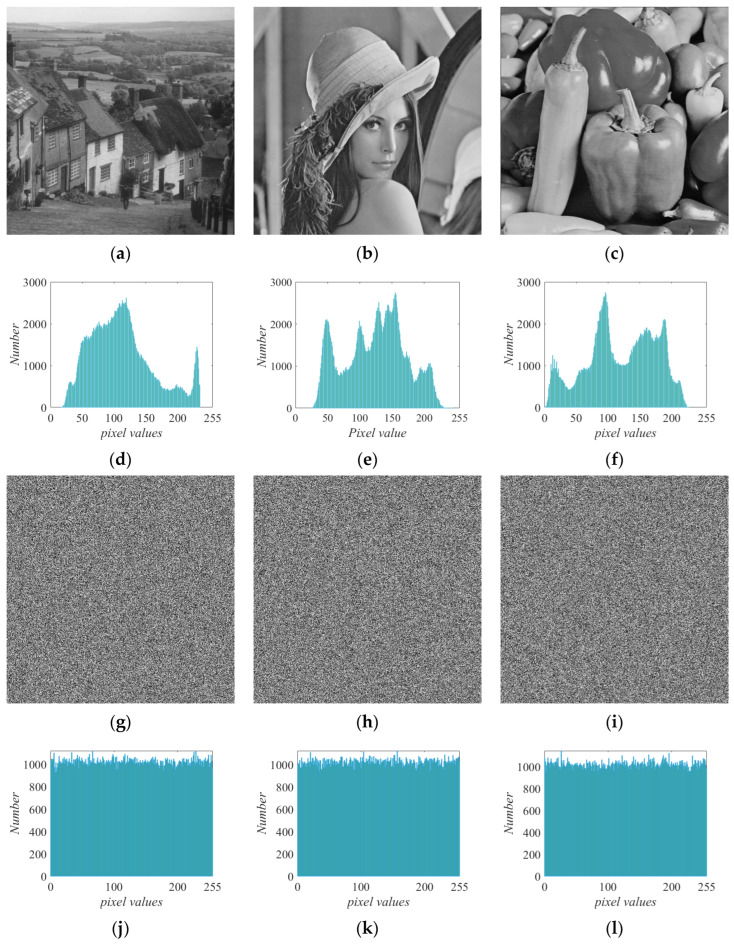
Encryption results of images Goldhill, Lena, and Pepper and their histograms. Plain images of (**a**) Goldhill, (**b**) Lena, and (**c**) Pepper; histograms of plain images of (**d**) Goldhill, (**e**) Lena, and (**f**) Pepper; ciphered images of (**g**) Goldhill, (**h**) Lena, and (**i**) Pepper; histograms of ciphered images of (**j**) Goldhill, (**k**) Lena; and (**l**) Pepper.

**Table 1 entropy-25-01656-t001:** Results of the NIST test.

Statistical Test	*p*-Value	Result
Monobit	0.5485	pass
Block Frequency	0.8862	pass
Runs	0.6310	pass
Longest Runs	0.6912	pass
Rank	0.0352	pass
DFT	0.7941	pass
Nonoverlapping Template	0.9754	pass
Overlapping Template	0.0623	pass
Universal	0.9118	pass
Linear Complexity	0.7493	pass
Serial	0.711	pass
Approximate Entropy	0.0345	pass
Cumulative Sums	0.945	pass
Random Excursions	0.454	pass
Random Excursions Variant	0.697	pass

**Table 2 entropy-25-01656-t002:** $\chi^{2}$ test.

Images	Goldhill	Lena	Pepper
plain	1.6162 × 10^5^	1.5835 × 10^5^	1.2016 × 10^5^
ciphered	256.7324	186.0684	243.4199

**Table 3 entropy-25-01656-t003:** Correlation coefficients of each image in each direction.

Test Image	Plain Image			Ciphered Image		
	Horizontal	Vertical	Diagonal	Horizontal	Vertical	Diagonal
Lena (proposed)	0.9845	0.97	0.9632	−0.0049	0.0095	−0.0106
Lena (ref. [21])	0.9850	0.9719	0.9576	0.0231	0.0138	−0.0002
Lena (ref. [22])	0.9854	0.9736	0.9552	−0.0084	0.0065	0.0036
Lena (ref. [23])	0.9673	0.9412	0.9132	0.0171	−0.0013	−0.0005
Lena (ref. [24])	0.97688	0.9835	0.9534	−0.0088	0.0048	0.0053
Lena (ref. [25])	0.9024	0.9287	0.8856	−0.0082	0.0094	0.0102
Goldhill	0.974	0.9739	0.9445	0.0049	−0.0069	−0.0158
Pepper	0.9813	0.9804	0.9676	−0.0143	−0.06	0.00249

**Table 4 entropy-25-01656-t004:** Information entropy of test images.

Test Image	Plain Image	Ciphered Image
Goldhill	7.4451	7.99936
Pepper	7.5937	7.99928
Pepper (ref. [20])	7.6698	7.9998
Lena (proposed)	7.4778	7.99933
Lena (ref. [27])	7.3982	7.9692
Lena (ref. [28])	7.5683	7.9978
Lena (ref. [29])	7.4451	7.9993

**Table 5 entropy-25-01656-t005:** Average local information entropy of test images.

Test Image	Plain Image	Ciphered Image
Goldhill	4.7113	0.0293
Lena	4.347	0.0282
Pepper	4.4384	0.0308

**Table 6 entropy-25-01656-t006:** Key sensitivity analysis results.

Indicator		Goldhill	Lena	Pepper	Theoretical Value
*X* _0_	NPCR (%)	99.6096	99.6098	99.6089	99.6094
	UACI (%)	33.4653	33.4649	33.4653	33.4635
	BACI (%)	26.7730	26.7714	26.7730	26.7712
*Y* _0_	NPCR (%)	99.6097	99.6099	99.6094	99.6094
	UACI (%)	33.4625	33.4654	33.4605	33.4635
	BACI (%)	26.7709	26.7709	26.7705	26.7712
*Z* _0_	NPCR (%)	99.6089	99.6095	99.6083	99.6094
	UACI (%)	33.4625	33.4652	33.4633	33.4635
	BACI (%)	26.7702	26.7709	26.7709	26.7712
*W* _0_	NPCR (%)	99.6099	99.6089	99.6099	99.6094
	UACI (%)	33.4640	33.4625	33.4640	33.4635
	BACI (%)	26.7698	26.7699	26.7798	26.7712

## Data Availability

The original contributions presented in the study are included in the article. Further inquiries can be directed to the corresponding author.

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
