# Peer review of "A Conservative Memristive Chaotic System with Extreme Multistability and Its Application in Image Encryption"

_entropy, 2023, doi:10.3390/e25121656_

Round 1

Reviewer 1 Report

Comments and Suggestions for Authors

Some comments as:

1)       Improve the abstract to point better to the scientific contribution and novelty of the proposal and insert already the abstract quantitative data that demonstrates the benefits of your approach.

2)       The presented image encryption algorithm is not clear and both diffusion and scrambling operations are not defined in this algorithm.

3)       The plain image sensitivity is not satisfied for the proposed encryption algorithm. Hint: https://doi.org/10.1016/j.jisa.2022.103367

4)       The decryption algorithm of the suggested cryptosystem is not stated.

5)       Lena's image is a crop from the November 1972 Playboy centerfold image, which is inappropriate for scientific publications. Use another image.

6)       In addition to histogram analysis, the chi-square test can be performed to validate the results statistically. Hint: https://doi.org/10.1007/s40747-023-00988-7

7)       Perform local entropy test in addition to global entropy.

8)       Noise and data loss attacks are needed to be tested.

9)       Time complexity analysis for image cryptosystem is not stated.

10)   Include the performance analysis phase and compare the proposed encryption algorithm to related published works.

11)   After comparing the methods, you have to be able to analyze the results and relate them to the structure of each algorithm. It would be interesting to have your thoughts on why the better algorithm works that way. Such analyses would be the core of your work where you prove your understanding of the reason behind the results. You can also link the findings to the hypotheses of the paper, which is to improve the exploration and exploitation of the proposed method for the many-objective problem investigated. Long story short, this paper requires very deep analysis from different perspectives.

12) The conclusion section must mention the limitations and future scope of this study.

Author Response

Thank you very much for taking the time to review this manuscript. The corresponding revisions in the re-submitted files. Please see the attachment.

Reviewer 2 Report

Comments and Suggestions for Authors

It is best to declare abbreviations when using them.

Eq. (6) should be wrong. The characteristic equation is not the same matrix J0.

In Section 3, the figure descriptions differ slightly from the manuscript descriptions.

Figure 7 should be divided into two types of figures.

It is best to declare abbreviations when using them.

In Subsection 5.2, it is an important result in this paper, which can also be represented by a flow chart. Each step should be clarified  in the previous sections. For example, how to perform the scrambling operation is not clearly explained.

Comments on the Quality of English Language

The paper should improve the English. For example, in the article, the font size should be consistent. The upper and lower case writing should also be consistent in the Section title, and the English usage should be double check. 

Author Response

(The authors gave the same response as above.)

Reviewer 3 Report

Comments and Suggestions for Authors

This paper investigates the multistability of a novel conservative memristive chaotic system and its applicaiton in image encryption. The results seem correct. The following comments need to be addressed in the revision. 

1. The motivation and the difficulties of the new memristive chaostic model need to be added. Morever, the superiorities of the new memristive chaotic system in image encryption application need to be enhanced with other related work. The current introduction need to to be enhanced.

2. The contribution part need to be added in introduction.

3. Lots of mistakes. For instance, the Where behind (1)(2)……, Where needs to to changed into where; the comma in the systems; the v is not consistent in (1) and in the paragraphy, and the same problem z(0)=u exists in 3.3.1.

4. The procedure of system (2) to system (3) is not clear. Why the first eauation of system (3) is not consistent with the one in (2)? What do you mean by internal state variable of the memristor?  Besides, the procedure of image encryption scheme is not clear, some basic equations are missed.

5. For image encryption scheme and applicaiton of memristive chaotic systems, the related work can be refered and compared as finite/fixed-time synchronization of memrisotr chaotic system and image encryption application and synchronization of a memritor chaotic system and image encryption.

6. The comparisons of the results as well as the encryption algrithm are better to be added.

7. The future work and the drawback are advised to be added.

8. The format of the references needs to be corrected.

Comments on the Quality of English Language

 Moderate editing of English language required and some mistakes need to be corrected.

Author Response

Thank you very much for taking the time to review this manuscript. The corresponding revisions in the re-submitted files. Please see the attachment。

Round 2

Reviewer 1 Report

Comments and Suggestions for Authors

The previous comments have been addressed.

Author Response

Thank you very much for your time in reviewing our manuscript.

Reviewer 2 Report

Comments and Suggestions for Authors

The authors should spend more time to organize the typesetting of the paper.

Eq (4) is not used in the article and can be deleted. And Eq. (6) still is wrong. 

In the section 4, the authors can build a flow chat to explain the process and methodology of the implementation based on STM32. The description in section 4 is short compared with the other section. 

Comments on the Quality of English Language

The authors should spend more time to organize the typesetting of the paper.

Author Response

Thank you very much for your time in reviewing our manuscript. Please find the detailed responses  in the re-submitted files. Please see the attachment.

Reviewer 3 Report

Comments and Suggestions for Authors

This revised version can be accepted.

Comments on the Quality of English Language

 Minor editing of English language required and the consistence of the references.

Author Response

Thank you very much for your time in reviewing our manuscript.We have double-checked the whole manuscript and edited a little English.

Round 3

Reviewer 2 Report

Comments and Suggestions for Authors

No.